# Fermented Soybean Paste Attenuates Biogenic Amine-Induced Liver Damage in Obese Mice

**DOI:** 10.3390/cells12050822

**Published:** 2023-03-06

**Authors:** Ju-Hwan Yang, Eun-Hye Byeon, Dawon Kang, Seong-Geun Hong, Jinsung Yang, Deok-Ryong Kim, Seung-Pil Yun, Sang-Won Park, Hyun-Joon Kim, Jae-Won Huh, So-Yong Kim, Young-Wan Kim, Dong-Kun Lee

**Affiliations:** 1Department of Physiology and Convergence Medical Science, Institute of Health Sciences, Gyeongsang National University Medical School, Jinju 52727, Republic of Korea; 2Department of Biochemistry and Convergence Medical Science, Institute of Health Sciences, Gyeongsang National University Medical School, Jinju 52727, Republic of Korea; 3Department of Pharmacology and Convergence Medical Science, Institute of Health Sciences, Gyeongsang National University Medical School, Jinju 52727, Republic of Korea; 4Department of Anatomy and Convergence Medical Science, Institute of Health Sciences, Gyeongsang National University Medical School, Jinju 52727, Republic of Korea; 5National Primate Research Center, Korea Research Institute of Bioscience and Biotechnology, Cheongju 28116, Republic of Korea; 6Fermented and Processed Food Science Division, National Institute of Agricultural Sciences, Wanju-Gun 55365, Republic of Korea; 7Department of Food Science and Biotechnology, Korea University, Sejong 30019, Republic of Korea

**Keywords:** biogenic amine, obesity, liver, NAFLD, IL-1β, MAO

## Abstract

Biogenic amines are cellular components produced by the decarboxylation of amino acids; however, excessive biogenic amine production causes adverse health problems. The relationship between hepatic damage and biogenic amine levels in nonalcoholic fatty liver disease (NAFLD) remains unclear. In this study, mice were fed a high-fat diet (HFD) for 10 weeks to induce obesity, presenting early-stage of NAFLD. We administered histamine (20 mg/kg) + tyramine (100 mg/kg) via oral gavage for 6 days to mice with HFD-induced early-stage NAFLD. The results showed that combined histamine and tyramine administration increased cleaved PARP-1 and IL-1β in the liver, as well as MAO-A, total MAO, CRP, and AST/ALT levels. In contrast, the survival rate decreased in HFD-induced NAFLD mice. Treatment with manufactured or traditional fermented soybean paste decreased biogenically elevated hepatic cleaved PARP-1 and IL-1β expression and blood plasma MAO-A, CRP, and AST/ALT levels in HFD-induced NAFLD mice. Additionally, the biogenic amine-induced reduction in survival rate was alleviated by fermented soybean paste in HFD-induced NAFLD mice. These results show that biogenic amine-induced liver damage can be exacerbated by obesity and may adversely affect life conservation. However, fermented soybean paste can reduce biogenic amine-induced liver damage in NAFLD mice. These results suggest a beneficial effect of fermented soybean paste on biogenic amine-induced liver damage and provide a new research perspective on the relationship between biogenic amines and obesity.

## 1. Introduction

Biogenic amines are biologically activated low-molecular-weight nitrogenous organic compounds that are primarily produced by spoilage microorganisms mediating the enzymatic decarboxylation of amino acids. Representative biogenic amines include the aliphatic compounds putrescine, cadaverine, agmatine, spermine, and spermidine; the aromatic compounds tyramine and 2-phenylethylamine; and the heterocyclic compounds histamine and tryptamine. Although the toxicity of small amounts of biogenic amines is negligible, consuming large amounts of aromatic and heterocyclic compounds in food can be hazardous and cause serious health problems [1]. Unlike the fermentation process occurring under certain conditions and in certain environments, food decomposition results in the elevation of biogenic amine levels in foods. A high concentration of biogenic amines can adversely affect the nervous and vascular systems and may cause physiologically harmful reactions or intoxication [1,2]. Furthermore, biogenic amines ingested in large quantities from foods can enter the systemic circulation and consequently cause migraine, elevation of blood sugar levels, high blood pressure, Parkinson’s disease, schizophrenia, and depression [3]. In particular, histamine, a representative biogenic amine present in most foods, can cause histamine and scombrid poisoning if consumed in large amounts. Additionally, tyramine is as common as histamine and is abundant in various foods, including strong/aged cheeses, aged/smoked meats, wine, and avocados [2,4]. Similar to histamine, high levels of tyramine intake can have adverse health effects [1,2]. Additionally, the association effect of histamine and tyramine shows synergistic cytotoxicity in intestinal cells [5].

In general, small amounts of ingested biogenic amines from foods are physiologically metabolized and converted to less active forms via the detoxifying enzymes monoamine oxidase (MAO) and diamine oxidase (DAO) [6]. The mitochondrial enzyme MAO has two isoforms (MAO-A and -B) that are widely expressed in various organs, including the brain, heart, lungs, kidney, intestine, and liver [3]. MAO plays a physiologically important role in the metabolism of monoaminergic neurotransmitters in the central nervous system and biogenic amines in peripheral tissues [7]. Tyramine is a well-known substrate for MAO-A and causes hypertension and even death when combined with MAO inhibitors [8,9]. Furthermore, histamine is metabolized in the liver and eliminated from the blood after it becomes inactive [10]. Because DAO is rarely expressed in the livers of most species, histamine N-methyltransferase, instead of DAO, converts histamine to N-methylhistamine in the liver and is then metabolized by MAO-B [11,12]. In several studies, MAO activity has been used to estimate the levels of biogenic amines, especially histamine and tyramine [1,13]. Therefore, MAO activity can be expected to be essential in reducing biogenic amines in vivo. However, there is insufficient evidence to establish a correlation between biogenic amines and hepatic MAO activity.

Obesity, which has been increasing worldwide for decades, causes various health problems, such as metabolic disorders. Excessive fat accumulation induced by obesity causes metabolic diseases, such as type 2 diabetes mellitus (T2DM). Obesity contributes to the development of fatty liver, leading to nonalcoholic fatty liver disease (NAFLD) [14]. Recent studies suggest that T2DM is a critical risk factor for NAFLD development [14,15]. Several studies have also demonstrated that chronic and progressive fatty liver conditions caused by obesity can lead to advanced fibrosis, cirrhosis, hepatocellular carcinoma, and liver-related death [16,17,18,19]. Additionally, the interleukin 1 (IL-1) family of cytokines plays a pivotal role in NAFLD development. For instance, IL-1α and -1β promote fatty liver disease processes, including liver steatosis, hepatic damage, liver fibrosis, and the recruitment of immune cells induced by inflammation through IL-1 receptor signaling [20,21]. Although there is evidence for the deleterious effect of histamine and tyramine on the liver, the adverse risk of the relationship between NAFLD and biogenic amines has not been established.

Recent studies have documented that soybean-derived foods such as fermented soybean paste contain various beneficial components. The long-term ingestion of fermented soybean paste prevents high-fat diet (HFD)-induced metabolic disorders, including NAFLD and insulin resistance. Consequently, it lowers the incidence of T2DM [22,23,24]. Therefore, the aim of this study was to demonstrate increased hepatic damage caused by biogenic amines and the therapeutic role of fermented soybean paste after exposure to biogenic amines in HFD-induced NAFLD.

## 2. Materials and Methods

### 2.1. Animal Experiments

All experimental and animal care protocols were approved by the Gyeongsang National University Institutional Animal Care and Use Committee (GNU IACUC, GNU-200820-M0053) and performed following the National Institute of Health (NIH) guidelines and a scientifically reviewed protocol (GLA-100917-M0093). C57BL/6 mice were used in these experiments. The mice were fed a high-fat (60%) diet (Research Diets, Inc., New Brunswick, NJ, USA) for 10 weeks after weaning to induce NAFLD.

### 2.2. Preparation of Fermented Soybean Paste Powder

The fermented soybean paste powder used in the animal experiments was selected based on the National Health and Nutrition Survey of the Ministry of Health and Welfare of Korea. It was collected from 15 types of traditionally fermented soybean paste and two types of factory-made products. After grinding, the fermented soybean paste samples were quantified using a sterilized container. Then, 1000 g of each of 15 types of traditional fermented soybean paste were mixed to produce a standard sample of 15 kg. Two factory-made fermented soybean pastes (7.5 kg each) were mixed to make a standard sample of 15 kg. Each sample was freeze-dried for 5 days, then pulverized. The prepared soybean paste powder was stored at −20 °C. Residual biogenic amines were not removed from the samples.

### 2.3. Measurement of Body Weight, Food Intake and Survival Rate

Mice were randomly assigned and fed either a normal chow diet (NCD) or an HFD containing 10% or 60% fat (Research Diets, Inc., New Brunswick, NJ, USA) for 10 weeks. Mouse body weights were measured daily during oral gavage administration. The food and water intake of the mice was measured at 12 h intervals during the last day of the experiment from 7 pm to 7 am and from 7 am to 7 pm. The survival/mortality of the mice was recorded after 6 days of oral gavage administration of drugs.

### 2.4. Drug Administration

Drugs for oral gavage administration, including histamine (histamine dihydrochloride, TCI, Tokyo, Japan) and tyramine (Cayman Chemical, Ann Arbor, MI, USA), were dissolved in 0.5% carboxymethylcellulose (CMC, Sigma-Aldrich, St. Louis, MO, USA). The soybean paste powder was administered orally (75 or 750 mg/kg) with histamine and tyramine. A total volume of 0.5% CMC was treated to avoid exceeding the recommended dose for mice (10 mL/kg).

### 2.5. ELISA Assay

Blood samples were collected from the hearts and stored in ethylene glycol tetra-acetic acid-coated tubes (Becton, Dickinson and Company, Franklin Lakes, NJ, USA). The blood samples were centrifuged at 3000 rpm for 10 min at 4 °C, and each sample’s supernatant (blood plasma) was collected. This process was performed twice to obtain clear blood plasma samples. The collected blood plasma was immediately used for the ELISA assay to avoid degradation effects on the results. The total MAO, MAO-A, -B, bile acids, and C-reactive protein (CRP) levels in blood plasma were determined using OxiSelected MONOAMINE OXIDASE ASSAY KIT (Cell Biolabs, Inc., San Diego, CA, USA), Mouse Total Bile Acids Kit, and Mouse C-Reactive Protein ELISA Kit (Crystal Chem, Elk Grove Village, IL, USA) according to the manufacturers’ instructions. The absorbance of the samples was measured using a Versamax microplate reader (Molecular Devices, LLC., San Jose, CA, USA). The blood concentration of each protein was calculated according to the manufacturer’s instructions.

### 2.6. Plasma Biochemical Assays

The blood samples were collected following the protocol described in the ELISA Assay section. Plasma aspartate aminotransferase (AST) and alanine aminotransferase (ALT) levels were measured using dedicated kits (IVD Lab, Uiwang, Republic of Korea) and a spectrophotometer (Shimadzu UV-1800 spectrophotometer, Tokyo, Japan).

### 2.7. Western Immunoblotting

Isolated liver samples were homogenized in RIPA buffer (Thermo Scientific, Rockford, IL, USA) on ice for 30 min and centrifuged twice at 13,000 rpm for 30 min at 4 °C. The concentrations of solubilized proteins in the supernatants were determined using a BCA protein assay (Thermo Scientific). Proteins in supernatants (10 μg) were separated using 10% sodium dodecyl sulfate-polyacrylamide gel electrophoresis. The separated proteins were transferred to a methanol-activated polyvinylidene difluoride membrane (Merck, Darmstadt, Germany). The membrane was blocked with a blocking buffer containing 5% skim milk in a mixture of Tris-buffered saline and 0.1% Tween-20 and washed three times for 10 min. Membranes were then probed with either a primary rabbit antiserum against IL-1β (1:1000, Abcam, Cambridge, MA, USA) or osteopontin (1:1000) (Abcam, Cambridge, MA, USA) or PARP-1 (1:1000, Cell Signaling Technology, Danvers, MA, USA) for 18 h at 4 °C, rewashed three times, and incubated with horseradish peroxidase-labeled goat anti-rabbit secondary antiserum (1:3000) (Thermo Fisher Scientific, Tewksbury, MA, USA) for 1 h at room temperature. Immunoreactive protein bands were detected using an iBright Western blot imaging system (Thermo Scientific, Tewksbury, MA, USA) with enhanced chemiluminescence reagents (Ab Frontier, Seoul, Republic of Korea; ratio of reagents A to B = 1:500). The same membrane was stripped and probed with mouse primary antiserum against β-actin (1:1000) (Sigma-Aldrich, St. Louis, MO, USA) to normalize the blots. Immunoreactive protein bands were semiquantified using a digital imaging camera and NIH Image 1.62 software.

### 2.8. IPGTT

After 16 h of fasting, glucose solution (2 mg/kg, i.p.) was administered to the mice. Blood glucose levels were measured at 0, 30, 60, 90, and 120 min using a glucose meter (MEDISENSOR, Daegu, Republic of Korea). A blood sample was collected from the tail vein of the mouse, and the first drop of blood was discarded. The area under the curve from the IPGTT was calculated using the trapezoidal rule.

### 2.9. Statistics

Statistical analyses were performed using one-way analysis of variance with Tukey’s multiple comparison test (GraphPad Prism 9.3.1, GraphPad Software, La Jolla, CA, USA). Data were considered significantly different when the *p*-value was <0.05. All statistical results are presented as mean ± SEM.

## 3. Results

### 3.1. Changes in Survival Rate and Plasma CRP Levels after Repeated Exposure to Combined Biogenic Amines in Mice Fed an NCD

The mice were administered combined biogenic amines once a day for 6 days by oral gavage to determine the adverse effects of biogenic amines, histamine, and tyramine under NCD-fed conditions (Figure 1A). The combined biogenic amine administration was determined at three concentration levels—low (2 mg/kg histamine + 10 mg/kg tyramine, *n* = 12), medium (20 mg/kg histamine + 100 mg/kg tyramine, *n* = 12), and high concentration (200 mg/kg histamine + 1000 mg/kg tyramine, *n* = 12)—and 0.5% CMC without biogenic amines was used as a control (*n* = 12). Three concentration levels of combined biogenic amines were used—low (2 mg/kg histamine + 10 mg/kg tyramine, *n* = 12), medium (20 mg/kg histamine + 100 mg/kg tyramine, *n* = 12), and high concentration (200 mg/kg histamine + 1000 mg/kg tyramine, *n* = 12)—and CMC (0.5%) without biogenic amines was used as a control (*n* = 12). Body weight and food intake (but not water intake) were significantly reduced after treatment with high concentrations of biogenic amines (Figure 1B–D). However, medium and low concentrations of combined biogenic amines did not affect body weight and food intake. The survival rate of mice was 100% (*n* = 26) in the CMC control group but decreased to 91.7% (1 death out of a total of 12 mice) in the low-concentration group, 84.6% (2 deaths out of a total of 13 mice) in the medium-concentration group, and 46.2% (7 deaths out of a total of 13 mice) in the high-concentration administration group. Subsequently, to determine the adverse effects of biogenic amines on liver damage, we tested changes in CRP levels in blood plasma, a marker protein produced by hepatocytes and associated with NAFLD and inflammation [25]. Blood plasma CRP levels were significantly increased in the medium- (42.7 ± 7.9 ng/mL, *n* = 10) and high-concentration (52.3 ± 9.7 ng/mL, *n* = 13) groups but not in the group administered a low concentration (24.3 ± 0.9 ng/mL, *n* = 15) of biogenic amines compared with the control group (14.2 ± 2.1 ng/mL, *n* = 9) (Figure 1E,F). Therefore, the optimal dose for combined biogenic amine administration was determined to be a medium concentration, which increased CRP levels without affecting feeding behavior and survival. When histamine and tyramine were administered alone, there was no change in the survival rate of the experimental animals; however, the blood plasma CRP level increased significantly after tyramine was administered alone (Figure 1G,H).

### 3.2. Changes in Liver IL-1β Expression Levels after Repeated Exposure to Biogenic Amines in Mice Fed an NCD

Recent studies demonstrate that IL-1β cytokine is closely associated with inflammation, hepatic injury, and obesity [26,27]. Osteopontin is also a potential biomarker for numerous liver diseases [28,29]. Therefore, we investigated whether biogenic amines affect the expression levels of IL-1β and osteopontin in the mouse liver. Administration of histamine or tyramine alone did not change the liver expression levels of IL-1β, but the levels were increased by combined biogenic amine administration in the liver (NCD + CMC: *n* = 6; NCD + histamine 20 mg/kg: *n* = 6, NCD + tyramine 100 mg/kg: *n* = 6; NCD + histamine 20 mg/kg + tyramine 100 mg/kg: *n* = 6) (Figure 2A). Osteopontin expression levels in the mouse liver also showed an increasing tendency with biogenic amine administration, but the difference was not statistically significant (NCD + CMC: *n* = 6; NCD + histamine 20 mg/kg: *n* = 6, NCD + tyramine 100 mg/kg: *n* = 5; NCD + histamine 20 mg/kg + tyramine 100 mg/kg: *n* = 6) (Figure 2B). Therefore, in subsequent experiments, we used IL-1β as a marker to evaluate the effects of biogenic amines and hepatic damage in HFD-induced obesity. The full-length whole Western blot images for Figure 2 are shown in Figure A1.

### 3.3. Establishment of HFD-Induced NAFLD to Elucidate Biogenic Amine-Induced Liver Damage in Obesity

Leptin resistance is defined by a reduced sensitivity or a failure in brain response to leptin. Decreased tissue sensitivity to leptin leads to obesity and is closely linked to insulin insensitivity [15]. Furthermore, leptin indicates a predisposition to metabolic disorders, including fatty liver diseases [30,31]. Therefore, preliminary monitoring of leptin resistance is necessary to evaluate the effect of biogenic amines on obesity and NAFLD development. A previous study reported that C57BL/6 mice fed an HFD for 10 weeks showed symptoms of NAFLD [32]. Therefore, all mice used in our experiment were fed an HFD for 10 weeks to establish NAFLD. Mice fed an HFD (*n* = 12) showed decreased glucose tolerance compared to the NCD-fed group (*n* = 11) (Figure 3A,B). Additionally, fasting plasma glucose and plasma leptin levels were increased in HFD-fed mice for 10 weeks (Figure 3C,D). These data demonstrate that HFD-induced obese mice developed leptin resistance. These results demonstrate that this method establishes a model suitable for evaluating liver damage caused by biogenic amines in NAFLD.

### 3.4. Changes in Survival Rate and Liver Damage Markers after Single or Combined Biogenic Amine Administration in HFD-Induced Developmental NAFLD

To determine the effect of biogenic amines on the liver of HFD-induced obese mice, we tested the survival rate of experimental mice, IL-1β expression levels in the liver tissue, and blood CRP levels after oral gavage administration of biogenic amines. Survival rates were reduced after administration of both biogenic amine alone and a mixture of biogenic amines (CMC: 0 deaths out of 9, 100%; histamine 20 mg/kg: 1 death out of 13, 92%; tyramine 100 mg/kg: 2 deaths out of 12, 83%; histamine 20 mg/kg + tyramine 100 mg/kg: 7 deaths out of 33, 79%) (Figure 4A). Consistent with this result, liver IL-1β levels increased after biogenic amine treatment (HFD + CMC: *n* = 6; HFD + histamine 20 mg/kg: *n* = 7; HFD + tyramine 100 mg/kg: *n* = 7; HFD + histamine 20 mg/kg + tyramine 100 mg/kg: *n* = 6) (Figure 4B). Additionally, the survival rate was slightly lower in the HFD-fed group compared to the NCD-fed group (from 85% to 79%) (Figure 4C). Blood CRP levels were significantly increased in both CMC (*n* = 8) and biogenic amine-administered groups (*n* = 11) after HFD was fed compared to the NCD-fed, CMC-treated group (*n* = 9) (Figure 4D). As biogenic amines are degraded by the biogenic amine-detoxifying enzyme MAO [6], we determined whether biogenic amines alter liver MAO levels after HFD-induced NAFLD. Liver MAO-A and total MAO levels were significantly increased by repeated combined biogenic amine administration compared to the NCD control group, but MAO-B levels did not change (NCD + CMC: *n* = 6; HFD + CMC: *n* = 6; HFD + histamine 20 mg/kg: *n* = 6; HFD + tyramine 100 mg/kg: *n* = 7; HFD + histamine 20 mg/kg + tyramine 100 mg/ kg: *n* = 6) (Figure 4E–G). Although bile acids are derived from hepatic cholesterol catabolism and are closely associated with NAFLD development [33], repeated administration of combined biogenic amines did not change the total bile acid levels in the blood plasma (NCD + CMC: *n* = 6; HFD + CMC: *n* = 10; HFD + histamine 20 mg/kg: *n* = 10; HFD + tyramine 100 mg/kg: *n* = 10; HFD + histamine 20 mg/kg + tyramine 100 mg/kg: *n* = 11) (Figure 4H). These results demonstrate that even in the early stages of NAFLD, ingested biogenic amines may be associated with fatty liver conditions and induce severe risks linked to death via excessive response to MAO. The full-length whole Western blot images corresponding Figure 4B are shown in Figure A2.

### 3.5. Fermented Soybean Paste Affects Changes in Survival after Combined Biogenic Amine Administration in HFD-Induced Developmental NAFLD

To determine the effect of fermented soybean pastes on biogenically induced liver damage in HFD-induced NAFLD, traditionally made fermented soybean paste (TSBP) and manufactured (factory-made) fermented soybean paste (MSBP) feeding were combined with biogenic amines. The decreased survival rate after combined biogenic amine administration was increased by both TSBP (HFD + histamine 20 mg/kg + tyramine 100 mg/kg + TSBP 75 mg/kg: 1 death out of 9, 89%; HFD + histamine 20 mg/kg + tyramine 100 mg/kg + TSBP 750 mg/kg: 0 deaths out of 10, 100%) and MSBP (HFD + histamine 20 mg/kg + tyramine 100 mg/kg + MSBP 75 mg/kg: 0 deaths out of 9, 100%; HFD + histamine 20 mg/kg + tyramine 100 mg/kg + MSBP 750 mg/kg: 0 deaths out of 9, 100%) in the early stage of NAFLD (Figure 5A,B). We also evaluated blood aspartate aminotransferase (AST) and alanine aminotransferase (ALT) levels to determine liver damage by biogenic amines in developmental NAFLD. As shown in Figure 5C,D, combined biogenic amines induced increased levels of ALT (HFD + CMC: *n* = 4; HFD + histamine 20 mg/kg + tyramine 100 mg/kg: *n* = 5; HFD + histamine 20 mg/kg + tyramine 100 mg/kg + TSBP 75 mg/kg: *n* = 5; HFD + histamine 20 mg/kg + tyramine 100 mg/kg + TSBP 750 mg/kg: *n* = 4; HFD + histamine 20 mg/kg + tyramine 100 mg/kg + MSBP 75 mg/kg: *n* = 6; HFD + histamine 20 mg/kg + tyramine 100 mg/kg + MSBP 750 mg/kg: *n* = 6), and AST levels (HFD + CMC: *n* = 4; HFD + histamine 20 mg/kg + tyramine 100 mg/kg: *n* = 6; HFD + histamine 20 mg/kg + tyramine 100 mg/kg + TSBP 75 mg/kg: *n* = 5; HFD + histamine 20 mg/kg + tyramine 100 mg/kg + TSBP 750 mg/kg: *n* = 5; HFD + histamine 20 mg/kg + tyramine 100 mg/kg + MSBP 75 mg/kg: *n* = 5; HFD + histamine 20 mg/kg + tyramine 100 mg/kg + MSBP 750 mg/kg: *n* = 5) were decreased by TBST and MSBP administration for 6 days. These data suggest that both TSBP and MSBP may be involved in reducing biogenic amine-induced toxic effects associated with developmental NAFLD.

### 3.6. Effects of Fermented Soybean Paste on Changes in Liver Damage Markers after Combined Biogenic Amine Administration in HFD-Induced Developmental NAFLD

We evaluated changes in liver damage markers to determine whether fermented soybean paste reduces biogenic amine-induced liver damage in HFD-induced NAFLD. Combined biogenic amine-elevated hepatic IL-1β expression levels in developmental NAFLD were significantly decreased by both TSBP and MSBP (*n* = 6 per group, Figure 6A). We also evaluated changes in cleaved PARP-1, known as a cellular stress sensor, in developmental NAFLD. As shown in Figure 6B, biogenic amine-induced cleaved PARP-1 expression levels were significantly decreased by both TSBP and MSBP (*n* = 6 per group). Additionally, blood CRP levels and biogenic amines upregulated by HFD were significantly downregulated by fermented soybean paste (NCD + CMC: *n* = 6; HFD + CMC: *n* = 6; HFD + histamine 20 mg/kg + tyramine 100 mg/kg: *n* = 11; HFD + histamine 20 mg/kg + tyramine 100 mg/kg + TSBP 75 mg/kg: *n* = 5; HFD + histamine 20 mg/kg + tyramine 100 mg/kg + TSBP 750 mg/kg: *n* = 6; HFD + histamine 20 mg/kg + tyramine 100 mg/ kg + MSBP 75 mg/kg: *n* = 6; HFD + histamine 20 mg/kg + tyramine 100 mg + MSBP 750 mg/kg: *n* = 6) (Figure 6C). In particular, HFD and biogenic amine-induced enhanced activity of MAO-A levels was reduced by the high dose of MSBP, while MAO-B and total MAO levels did not change in the liver (NCD + CMC: *n* = 6; HFD + CMC: *n* = 6; HFD + histamine 20 mg/kg + tyramine 100 mg/kg: *n* = 6; HFD + histamine 20 mg/kg + tyramine 100 mg/kg + TSBP 75 mg/kg: *n* = 6; HFD + histamine 20 mg/kg + tyramine 100 mg/kg + TSBP 750 mg/kg: *n* = 6; HFD + histamine 20 mg/kg + tyramine 100 mg/kg + MSBP 75 mg/kg: *n* = 5; HFD + histamine 20 mg/kg + tyramine 100 mg/kg + MSBP 750 mg/kg: *n* = 6) (Figure 6D–F). These results support the hypothesis that fermented soybean paste reduces biogenic amine-enhanced hepatic damage in developmental NAFLD. The full-length whole Western blot images corresponding to Figure 6A,B are shown in Figure A3 and Figure A4, respectively.

## 4. Discussion

Although we are constantly exposed to the risk of biogenic amines by ingesting food with high protein or free amino acid contents, the relationship between biogenic amines and obesity-induced metabolic diseases remains elusive. Biogenic amines generated from fermentation decomposition by microorganisms or biochemical activity, including that histamine, tyramine, agmatine, putrescine, cadaverine, spermine, and spermidine, are not only toxic but also act as a measure of the freshness of food and spoilage [2]. Aliphatic biogenic amines are commonly used as a decay indicator. Aromatic and heterocyclic compounds act as ‘vasoactive amines’, causing toxicity by stimulating the nervous and vascular systems when consumed in excess [1,2].

The results of the present study show that ingesting large amounts of concomitant histamine and tyramine can induce life-threatening health problems. For instance, repeated exposure to combined biogenic amines decreased the food intake, survival rate, and body weight of NCD-fed mice. Additionally, repeated exposure to combined biogenic amines increased blood CRP levels in a dose-dependent manner. Immunoreactivity of the hepatic damage marker IL-1β, a well-known indicator of liver damage [21,34,35], was increased by the combined administration of biogenic amines. In particular, blood CRP levels increased only in HFD-induced NAFLD. Combined biogenic amine administration increased hepatic IL-1β levels in NAFLD. In contrast, survival rates were decreased by combined biogenic amine administration under normal conditions and showed a tendency to further reduce the survival rate in NAFLD mice. As IL-1β is closely associated with obesity and inflammatory fatty liver disease [21,34,35], HFD-induced NAFLD may be a factor in enhancing the risk of biogenic amines. IL-1β expression levels were increased, but developmental liver damage and the fibrogenesis marker osteopontin did not change. Therefore, these data suggest that the interaction between obesity-related NAFLD and unexpected ingestion of a large amount of biogenic amines may exacerbate hepatic function directly related to the maintenance of life.

The main risk factors for developing NAFLD are obesity, T2DM, and other factors associated with metabolic syndrome [36]. In general, NAFLD is induced by long-term HFD exposure, although a recent study reported that NAFLD symptoms appeared when after 10 weeks on an HFD [32]. However, the relationship between obesity factors and biogenic amines in the early stages of fatty liver disease is unclear. We monitored glucose and leptin levels as indicators of obesity due to the provision of HFD for 10 weeks. Glucose and leptin levels in blood plasma were significantly increased by 10 weeks of HFD feeding compared to those in the NCD-fed group (Figure 3). Therefore, we used this as a model for developmental NAFLD because HFD-mediated metabolic processes involved in leptin resistance accelerate de novo lipogenesis, inflammation, and fibrogenesis in the liver and consequently cause NAFLD [30,37].

Several studies have shown that IL-1β and CRP are strong predictors of NAFLD [21,33,38]. Hepatic IL-1β expression and blood CRP concentration were increased in the HFD-induced NAFLD group after combined biogenic amine administration compared with HFD + CMC and NCD + CMC groups (Figure 4). Similarly, the survival rate of the combined biogenic amine-treated group was lower that of the control group. CRP is produced by hepatocytes and is involved in chronic liver disease. Recent studies have shown that high CRP levels have been observed in patients with liver dysfunction [39,40]. In particular, patients with liver cancer or cirrhosis with high CRP levels show poor prognoses [41,42,43,44]. Additionally, IL-1β plays a critical role in hepatic failure via NF-κB signaling and proinflammatory cytokine activation [27]. These findings suggest that obesity may interact with repeated conjugated biogenic amine administration to cause liver damage via IL-1β and/or CRP upregulation.

It has been documented that NAFLD is closely associated with the upregulated activity of MAO-A in the liver, which is associated with oxidative stress-mediated depressive symptoms [45,46,47]. In addition, MAO inhibitors can potentially decrease the gene and protein expression of the proinflammatory cytokines IL-1β, IL-6, TNF-α, and INF-γ [3,45]. Consistent with these findings, we found that hepatic MAO-A, total MAO levels, and IL-1β expression increased after combined biogenic amine administration in HFD-fed NAFLD mice, while blood bile acid levels did not change significantly. Although the changes in MAO-B levels were not significant, they did show an increasing trend. Therefore, these findings suggest that hepatic MAO-A and IL-1β may act as more helpful markers than total bile acid when evaluating the risk of biogenic amine-induced liver damage in obese mice. In addition, recent studies demonstrated that cleavage of PARP-1, as a necrotic cell death marker, is closely associated with NAFLD and oxidated stress-induced liver damage [48,49].

Interestingly, fermented soybean paste recovered the survival rate reduced by biogenic amines in NAFLD **(Figure 5**). Usually, hepatic steatosis due to NAFLD causes increased ALT and AST levels [50,51]. Since plasma ALT and AST levels are known indicators of hepatocyte damage, they are used as general clinical biomarkers to evaluate liver function [52]. As shown in Figure 5, it was confirmed that increased blood ALT and AST levels by combined biogenic amine treatment were decreased in the TSBP- and MSBP-treated groups of NAFLD mice. Biogenic amine-induced increased IL-1β, cleaved PARP-1 expression levels, and blood CRP were decreased by concomitant administration of fermented soybean paste in NAFLD (Figure 6A–C). Additionally, fermented soybean paste reduced biogenic amine-enhanced hepatic MAO-A activity, but MAO-B and total MAO activities did not change (Figure 6D–F). These findings suggest that fermented soybean paste may relieve hepatic damage by reducing MAO, IL-1β, and cleaved PARP-1 levels increased by biogenic amines in HFD-induced developmental NAFLD.

Several studies have reported that fermented soybeans provide various benefits, including antioxidative, anti-inflammatory, fibrinolytic, anticancer, and immune-enhancing effects [53,54]. These benefits are known to be caused by isoflavones or secondary metabolites produced by microorganisms involved in the fermentation process [23,53,54,55,56]. Although decarboxylase-containing microorganisms produce biogenic amines during soybean paste fermentation, certain microorganisms, such as lactic acid bacteria and amine oxidase gene-containing bacteria, reduce biogenic amines [55,57,58]. Among the microbes, *Pediococcus acidilactici* M28 and *Staphylococcus carnosus* M43, which are contained in Chinese soybean paste, can degrade biogenic amines, especially tyramine and histamine [59]. Additionally, *Staphylococcus* is one a dominant microbe in fermented Korean soybean paste [60]. Therefore, it is feasible that fermented soybean paste can reduce biogenic amine-mediated liver damage in NAFLD patients. However, the detoxification mechanism of biogenic amines by certain microorganisms remains unclear but may occur in a hepatic MAO-independent manner (Figure 7).

Controlling spoilage microorganisms during fermentation is challenging because the production method varies from household to household and region to region. In contrast, systematically produced MSBP may have a constant amount or concentration of beneficial bacteria containing amine oxidase. Therefore, MSBP may be more effective in improving the liver damage caused by NAFLD biogenic amines. In this study, we did not investigate the effects of biogenic amines produced in soybean paste or the adverse effect of other biogenic amines except for histamine and tyramine. Nevertheless, recent studies demonstrate that the microbial community of soybean paste is involved in the degradation of other biogenic amines in addition to histamine and tyramine [55,57,58]. Although the precise mechanism of the beneficial effect of fermented soybean pastes on biogenic amine-induced liver damage remains unclear, our data suggest that fermented soybean pastes may contribute to a reduction in biogenic amine-induced liver damage in NAFLD.

## 5. Conclusions

We identified an increased risk of hepatic dysfunction by biogenic amine ingestion in obesity and confirmed that both blood and liver biomarkers, including CRP, MAO, IL-1β, and PARP-1, are helpful markers for evaluating hepatic function in biogenic amine-induced liver damage in obesity. Although there is insufficient documentation of the complementary benefits of soybean paste and the increased risk of biogenic amines in obesity, we propose that fermented soybean paste is a promising candidate to alleviate liver damage caused by biogenic amines in NAFLD.

## Figures and Tables

**Figure 1 cells-12-00822-f001:**
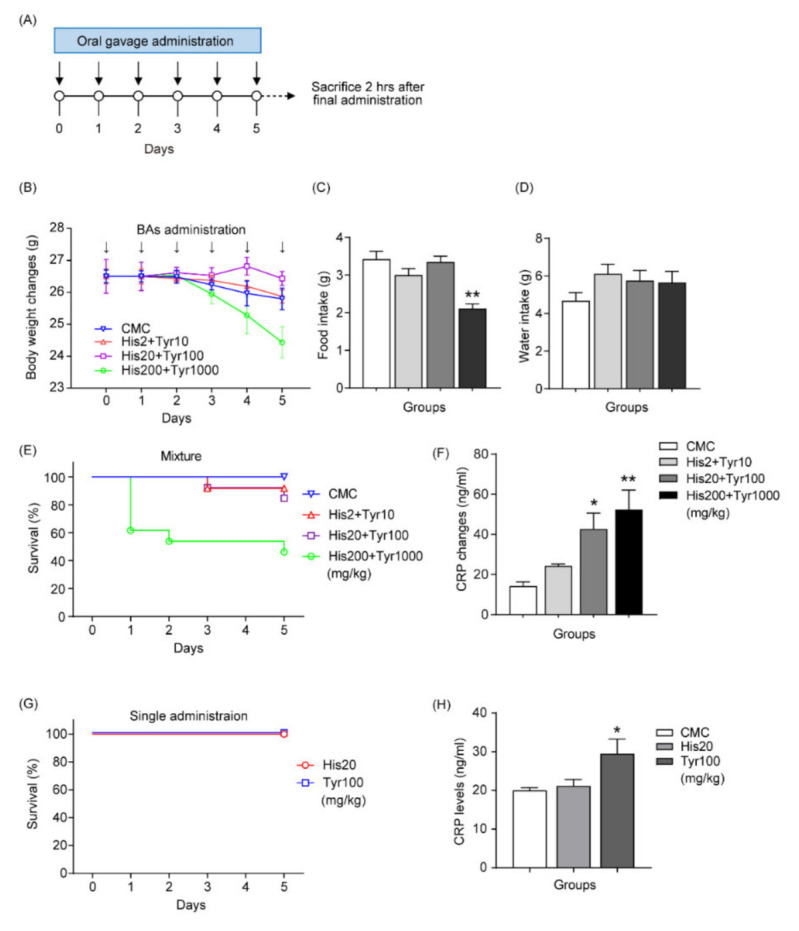
Combined effect of biogenic amines administered by repeated oral gavage in mice fed a normal chow diet (NCD). (**A**) Timeline of repeated oral gavage administrations of biogenic amines. Changes in body weight (**B**), food intake (**C**), and water intake (**D**) after repeated oral gavage administrations of biogenic amines. (**E**) Changes in survival rate after repeated oral gavage administrations of biogenic amines by concentration. (**F**) Changes in plasma C-reactive protein (CRP) levels after repeated oral gavage administrations of biogenic amines by concentration. (**G**) Changes in survival rate following repeated oral gavage administration of histamine and tyramine. (**H**) Changes in plasma CRP levels after repeated oral gavage administrations of either histamine or tyramine. * *p* < 0.05, ** *p* < 0.01 vs. carboxymethylcellulose (CMC). Data are shown as mean ± SEM. BA, biogenic amine; His, histamine; Tyr, tyramine.

**Figure 2 cells-12-00822-f002:**
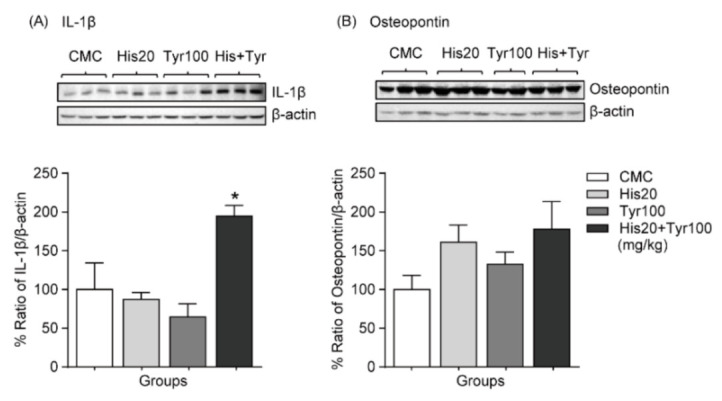
Effect of administration of single biogenic amines or their mixture on liver damage markers in mice fed an NCD. (**A**) Changes in IL-1β expression levels after repeated oral gavage administrations of histamine, tyramine, and combined biogenic amines. (**B**) Osteopontin expression levels changed after oral gavage administrations of histamine, tyramine, and combined biogenic amines. * *p* < 0.05 vs. CMC. Data are shown as mean ± SEM. IL-1β, interleukin-1 beta.

**Figure 3 cells-12-00822-f003:**
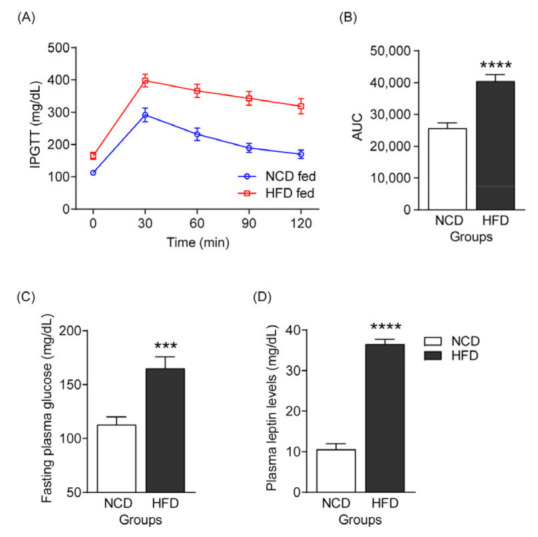
Establishing an HFD-induced NAFLD mice model by evaluating leptin resistance. (**A**) Intraperitoneal glucose tolerance test to evaluate leptin resistance after 10 weeks of feeding mice either an NCD or HFD. (**B**) The area under the curve corresponding to Figure 3A. (**C**) Fasting plasma glucose levels. (**D**) Plasma leptin levels. *** *p* < 0.001, **** *p* < 0.0001 vs. CMC. Data are shown as mean ± SEM.

**Figure 4 cells-12-00822-f004:**
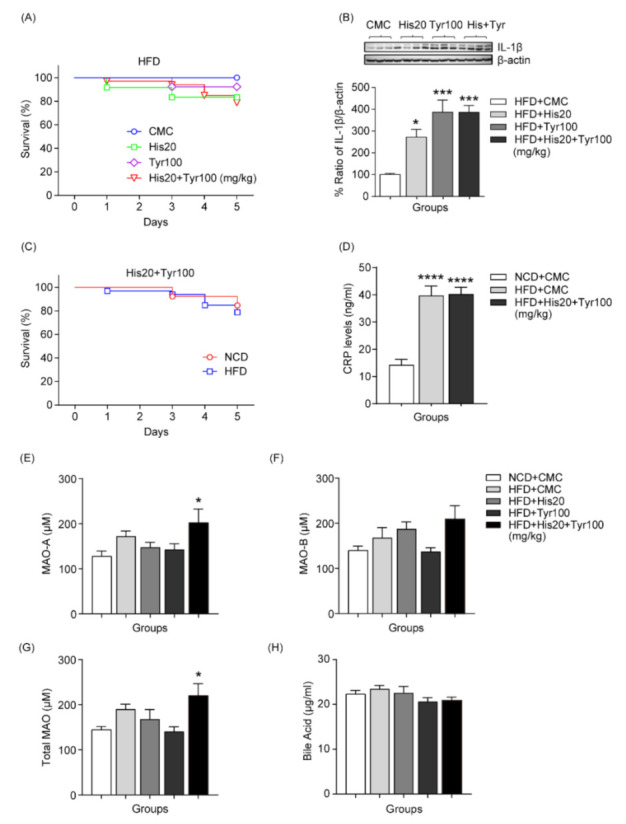
Effect of biogenic amines on liver damage in mice fed an HFD. (**A**) Changes in survival rate following single or repeated oral gavage administrations of biogenic amines after feeding an HFD. (**B**) Changes in IL-1β following single or repeated oral gavage administrations of biogenic amines after feeding an HFD. (**C**) Comparison of changes in survival rate between NCD- and HFD-fed groups after repeated oral gavage administrations of biogenic amines. (**D**) Comparison of changes in blood CRP levels between NCD-fed, HFD-fed, and HFD-fed + biogenic amines administration groups. Changes in liver MAO-A (**E**), MAO-B (**F**), total MAO (**G**) levels, and total bile acid levels in the blood (**H**). * *p* < 0.05, *** *p* < 0.001, **** *p* < 0.0001 vs. NCD + CMC. Data are shown as mean ± SEM. MAO, monoamine oxidase.

**Figure 5 cells-12-00822-f005:**
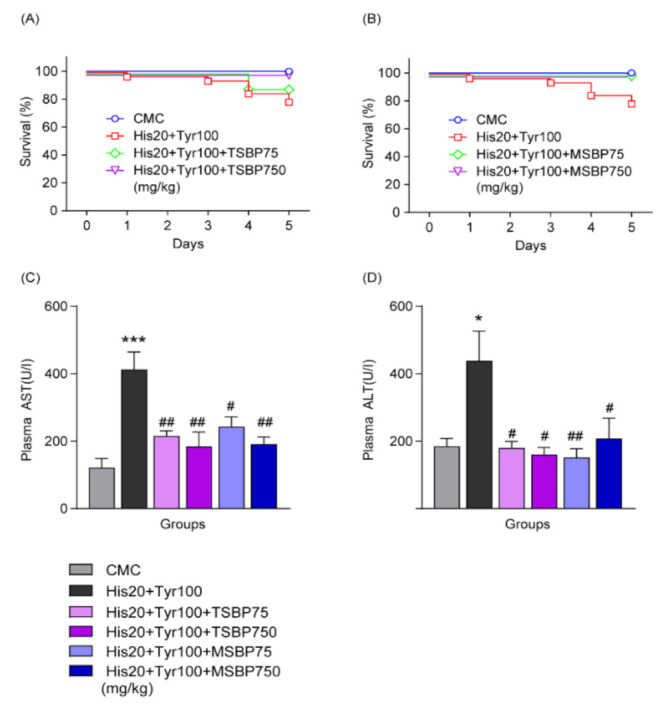
Reduction in biogenic amine-induced toxic effects by fermented soybean paste in obesity. (**A**) Effect of TSBP on survival rate changes caused by biogenic amine administrations and HFD-induced obesity. (**B**) Effect of MSBP on survival rate changes caused by biogenic amine administrations and HFD-induced obesity. (**C**) Changes in blood AST and (**D**) ALT levels caused by biogenic amines and fermented soybean paste in HFD-induced obesity. * *p* < 0.05, *** *p* < 0.001 vs. HFD + CMC; # *p* < 0.05, ## *p* < 0.01 vs. HFD + histamine 20 mg/kg + tyramine 100 mg/kg. Data are shown as mean ± SEM. TSBP, traditionally made fermented soybean paste; MSBP, manufactured fermented soybean paste.

**Figure 6 cells-12-00822-f006:**
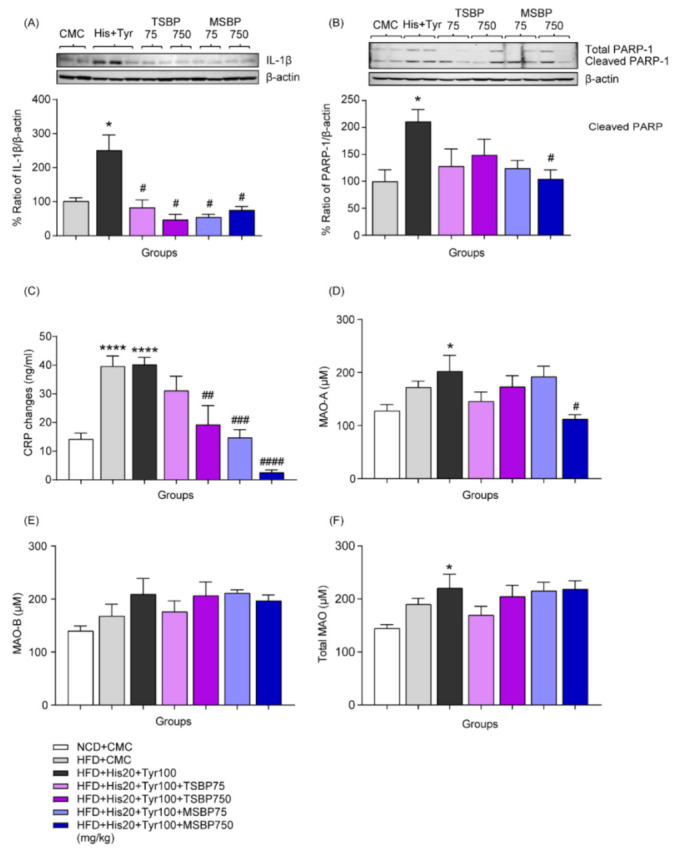
Reduction in biogenic amine-induced hepatic damage by fermented soybean paste in developmental NAFLD. (**A**) Changes in IL-1β expression levels by biogenic amines and fermented soybean paste in HFD-induced NAFLD liver tissue. (**B**) Changes in cleaved PARP-1 expression levels caused by biogenic amines and fermented soybean paste in HFD-induced NAFLD liver tissue. (**C**) Changes in blood CRP levels caused by biogenic amines and fermented soybean paste in HFD-induced NAFLD. Changes in MAO-A (**D**), MAO-B (**E**), and total MAO (**F**) levels caused by biogenic amines and fermented soybean paste in HFD-induced NAFLD liver tissue. * *p* < 0.05, **** *p* < 0.0001, vs. NCD + CMC; # *p* < 0.05, ## *p* < 0.01, ### *p* < 0.001, #### *p* < 0.0001, vs. HFD + histamine 20 mg/kg + tyramine 100 mg/kg. Data are shown as mean ± SEM.

**Figure 7 cells-12-00822-f007:**
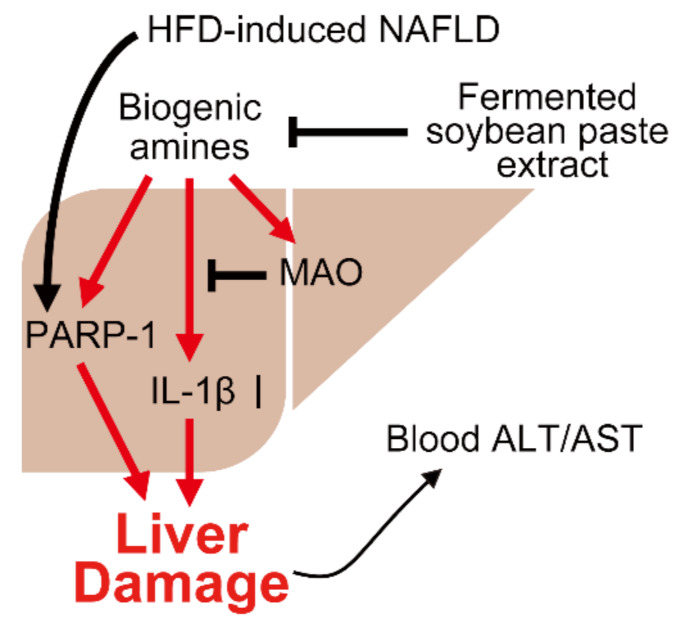
Putative role of fermented soybean paste extract in biogenic amine-induced hepatic damage in NAFLD. A large amount of combined biogenic amine ingestion may exacerbate hepatic function by increasing IL-1β expression, although activation of MAO degrades biogenic amines in NAFLD. In addition, biogenic amines enhance the cleavage of PARP-1, which may be upregulated by fatty liver disease. However, fermented soybean paste extracts are probably involved in the degradation of biogenic amines, reducing biogenic amine-induced hepatic damage in NAFLD.

## Data Availability

Not applicable.

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
