# Peer review of "Fermented Soybean Paste Attenuates Biogenic Amine-Induced Liver Damage in Obese Mice"

_cells, 2023, doi:10.3390/cells12050822_

Round 1
Reviewer 1 Report
This study described the effects of biogenic amines, combined histamine and tyramine, on deteriorating the liver damage in high fat diet-induced obese mice, and meanwhile the protective effect of fermented soybean paste was evaluated. Administration of biogenic amines decreased the mice survival rate under HFD condition, which was improved by soybean paste treatment. The following limitations were observed that diminished reviewer enthusiasm:
Major comments:
1) High histamine accumulation has been known to affect the nervous or vascular systems, leading to respiratory failure [PMID: 32587916], which might be the major cause of death for the histamine-treated mice in this study. To demonstrate the effect of biogenic amines on liver damage, the authors selected IL-1β as the biomarker, however, the evidence was too thin to make the conclusion that liver function was damaged after repeated exposure to biogenic amines. The increased IL-1β is insufficient to induce the death.
2) The kidneys have a considerable capacity for removing histamine. When healthy individuals were infused with histamine, a large proportion of the amine was methylated by the kidneys and excreted in urine [PMID: 32587916]. The function of liver for the metabolism of orally administrated histamine and tyramine is not illustrated.
3) The earliest stage of NAFLD, induced by 10-week HFD stimulation in this study, is hepatic steatosis characterized by simply fat accumulation in the liver with no or very low inflammation. The only one piece of data for IL-1β change without liver function determination is not solid enough to demonstrate the liver damage. In addition, as shown in Figure 4C, the survival rate was not significantly changed in the HFD-fed group compared to the NCD-fed group, suggesting that high level of biogenic amines-induced death is obesity-independent. Thus, the HFD model seems of little importance for this study.
4) This study described the reduction of biogenic amines-induced toxic effects by fermented soybean paste in obesity, the potential mechanism, however, is not given. Although several studies have reported that fermented soybeans provide various benefits, including antioxidative, anti-inflammatory, fibrinolytic, anti-cancer, and immune-enhancing effects, whether these benefits are involved in the biogenic amines-induced toxic model is still unknown. Additionally, the mice in one treatment group should receive 750 mg/kg fermented soybean paste + 20 mg/kg histamine + 100 mg/kg tyramine by gavage at a single dosage, but the volume of administration was not provided, which might exceed the maximum capacity tolerated by gavage in mice.
5) Since fermented soybean paste itself also contains high levels of histamine and tyramine, and the residual biogenic amines were not removed from the samples in this study, it’s not rigorous that the residual biogenic amines might affect the concentrations of administrated biogenic amines.
6) Fifteen types of traditional fermented soybean pastes or two factory-made fermented soybean pastes were mixed to produce a standard sample for this study. Since every type might underwent different spoilage microorganisms during fermentation, it’s hard to tell out the beneficial components.
Author Response
Reviewer 1
Comments and Suggestions for Authors
This study described the effects of biogenic amines, combined histamine and tyramine, on deteriorating the liver damage in high fat diet-induced obese mice, and meanwhile the protective effect of fermented soybean paste was evaluated. Administration of biogenic amines decreased the mice survival rate under HFD condition, which was improved by soybean paste treatment. The following limitations were observed that diminished reviewer enthusiasm:
Major comments:
1) High histamine accumulation has been known to affect the nervous or vascular systems, leading to respiratory failure [PMID: 32587916], which might be the major cause of death for the histamine-treated mice in this study. To demonstrate the effect of biogenic amines on liver damage, the authors selected IL-1β as the biomarker, however, the evidence was too thin to make the conclusion that liver function was damaged after repeated exposure to biogenic amines. The increased IL-1β is insufficient to induce the death.
Response: Thank you for your valuable critique. High histamine toxicity-induced respiratory failure is a well-known mechanism of the biogenic amine-induced major cause of death by histamine administration. However, respiratory failure might not be the only cause of death in mice caused by biogenic amines, and liver damage, which is responsible for detoxification, may also cause death. In this study, we wanted to show synergistic liver damage caused by the combination of biogenic amine consumption in obese condition. We already used IL-1β as a liver damage marker but, we added results for a hallmark for apoptosis, PARP-1, accoring your opinion [PMID: 35039483]. The data for cleaved PARP-1 expression levels in the liver showed in Fig 6B and explained on page 10, lines 355-358. Also, we added serum ALT/AST data, which are influential hepatic damage makers in Fig 5C,D and explained on page 9, lines 327-339.
2) The kidneys have a considerable capacity for removing histamine. When healthy individuals were infused with histamine, a large proportion of the amine was methylated by the kidneys and excreted in urine [PMID: 32587916]. The function of liver for the metabolism of orally administrated histamine and tyramine is not illustrated.
Response: It is hardly to find studies about liver damage and its mechanisms by exogenous supply of bioamines, therefore, we tried to figure out in this study that bioamines can cause liver damage, and we also tried to demonstrated that liver damage can lead to the early stages of NAFLD. Therefore, in this study, we discussed the effect of biological amines on the liver but not the kidney.
3) The earliest stage of NAFLD, induced by 10-week HFD stimulation in this study, is hepatic steatosis characterized by simply fat accumulation in the liver with no or very low inflammation. The only one piece of data for IL-1β change without liver function determination is not solid enough to demonstrate the liver damage. In addition, as shown in Figure 4C, the survival rate was not significantly changed in the HFD-fed group compared to the NCD-fed group, suggesting that high level of biogenic amines-induced death is obesity-independent. Thus, the HFD model seems of little importance for this study.
Response: We agree with your opinion about the early stage of NAFLD. The reviewer might be know that the effect of fermented soybean paste cannot outperform those of drugs. In this study, we only considered the effects of biogenic amine on the liver damage associated with the early stage of NAFLD and the beneficial effect of fermented soybean paste. The differences in mortality between NCD and HFD due to biogenic amine may be insignificant, but in the early stage of NAFLD, biogenic amines increased the IL-1β exprssion as a liver damage marker, as shown in Fig 4B. Also, we also added cleaved PARP-1 expression in the liver. Please refer to response #1. In fact, the effect of soybean paste on severe fatty liver disease is thought to be less or non effectivity. Therefore, we considered that the effect of fermented soybean paste would be better applied to an early obesity model. Taken these points together, we tried to explain that the harmful effects of biogenic amines on the liver in obesity and the beneficial effect of fermented soybean paste to alleviate liver damage in this study.
4) This study described the reduction of biogenic amines-induced toxic effects by fermented soybean paste in obesity, the potential mechanism, however, is not given. Although several studies have reported that fermented soybeans provide various benefits, including antioxidative, anti-inflammatory, fibrinolytic, anti-cancer, and immune-enhancing effects, whether these benefits are involved in the biogenic amines-induced toxic model is still unknown. Additionally, the mice in one treatment group should receive 750 mg/kg fermented soybean paste + 20 mg/kg histamine + 100 mg/kg tyramine by gavage at a single dosage, but the volume of administration was not provided, which might exceed the maximum capacity tolerated by gavage in mice.
Response: Since our study is focused on the liver damage caused by biogenic amines, the mechanism study of the beneficial mechanism of soybean paste has left as the further study topic. Sorry for the missing sentence. We used 0.5% CMC as a solvent to avoid exceeding the recommended dose for mice (10 ml/kg). We added the explanation sentence on page 3 line 131-137 in materials and methods section.
5) Since fermented soybean paste itself also contains high levels of histamine and tyramine, and the residual biogenic amines were not removed from the samples in this study, it’s not rigorous that the residual biogenic amines might affect the concentrations of administrated biogenic amines.
Response. As the reviewer pointed out, we did not remove the biogenic amines in the fermented soybean pastes. However, we considered that the concentration of orally administered biogenic amine was not very high and that the general concentration of biogenic amine contained in commercial soybean paste was low under the manufacturer's quality control. Also, as described in the material and method section, the soybean paste powder used in this study was selected based on the National Health and Nutrition Survey of the Ministry of Health and Welfare of Korea, so it can be seen as having public confidence. As shown in Fig 5 and 6, we confirmed that combined biogenic amine-induced increase of the hepatic IL-1β and PARP-1 expression and the plasma AST/ALT levels were decreased by fermented soybean paste treatment. Also, as shown in Fig. 5, when the fermented soybean paste was administered to the mice, it was confirmed that the survival rate of the mice improved. Therefore, we believe that fermented soybean paste without removing biogenic amines is also suitable to prove our hypothesis in this study.
6) Fifteen types of traditional fermented soybean pastes or two factory-made fermented soybean pastes were mixed to produce a standard sample for this study. Since every type might underwent different spoilage microorganisms during fermentation, it’s hard to tell out the beneficial components.
Response 1: Based on the data from Korea National Health and Nutrition Examination Survey, we considered representative and frequently consumed soybean paste as a treated sample. For that reason, we selected 2 kinds of factory-made and 15 kinds of traditionally-made soybean paste for this study.
Response 2: We did not analyze the ingredients of fermented soybean paste because the goal of this study was not to identify the specific mechanism or composition of fermented soybean paste to alleviate liver damage. Therefore, as the reviewer pointed out, we did not know the beneficial components of fermented soybean paste. However, our unpublished data showed that the reduction of increased biogenic amines during the fermentation process by certain beneficial lactic acid bacteria in the soybean paste samples. In addition, we already described in the discussion section, previous studies demonstrated that certain lactic acid bacteria and amine oxidase gene-containing bacteria reduce biogenic amines during fermentation in the soybean paste [PMID: 34925777; PMID: 26165318; PMID: 30823593]. Therefore, the freeze-dried soybean paste powder we used possibly contains various beneficial components. Taken together, we also believe that it is necessary to analyze the components of fermented soybean paste and specific mechanism of them that improves liver damage, it will be conducted as a follow-up study.
Reviewer 2 Report
In this study the authors demonstrate that fermented soybeans protect against the toxic effects of biogenic amines on the liver. Although this result deserves to be published, experiments on the mechanism of action and on the identification of compounds responsible for the protective effect are needed to elicit the interests of Cells readers.
In administration experiments why is carboxymethylcellulose administered in control animals and not amino acids from which biogenic amines are derived?
Author Response
Reviewer 2
Comments and Suggestions for Authors
In this study the authors demonstrate that fermented soybeans protect against the toxic effects of biogenic amines on the liver. Although this result deserves to be published, experiments on the mechanism of action and on the identification of compounds responsible for the protective effect are needed to elicit the interests of Cells readers.
In administration experiments why is carboxymethylcellulose administered in control animals and not amino acids from which biogenic amines are derived?
Response. We appreciate your positive comment. Thank you very much for your constructive advice. Because the tyramine powder is insoluble in water, we used 0.5% carboxymethylcellulose suspension for oral gavage administration. Therefore, we used 0.5 % CMC as a vehicle for control animals.
Round 2
Reviewer 1 Report
I appreciate the effort put in by the authors to improve the manuscript, but limitations of this study are suggested to be added to provide as potential reference for others' studies.
Author Response
We already described about limitation of our study in the discussion section. As following reviewer’s opinion, we added more description in the discussion section. Please refer to sentences on page 12 line 469-473.
Reviewer 2 Report
The insolubility of tyramine cannot justify the use of cellulose in control rats. This choice does not allow to appreciate the extent of the beneficial effects that are certainly present
The insolubility of tyramine cannot justify the use of carboxymetlcellulose in control rats. This choice does not allow to appreciate the extent of the beneficial effects that are certainly present
adding one experiment of control in which aminoacids and caroxycellulose was compared
Author Response
: This study is not about carboxymethylcellulose (CMC), but the effects of fermented soybean paste which alleviates liver dysfunction induced by exogenous biogenic amines, especially histamine and/or tyramine. We tried to induce liver damage by using histamine and tyramine. We had to dissolve histamine and tyramine to make them available for oral gavage administration. As we described first revision, since tyramine is not dissolved in water, the CMC was used to make biogenic amines suspension, and the CMC was used in all experimental groups, including the control group, not to consider the effect of CMC on the experimental results.
It is well known that the 2% CMC is a vehicle that is popularly used for oral administration in animal study [PMCID: 9655468]. In this study, a much lower concentration of 0.5% CMC was used, and only CMC was orally administered to the control group, so we believe that the effect of CMC did not affect the experimental results.
Since this study is not about the effect of each biogenic amine on liver damage, we believed that the comparison of other amino acids was not necessary to perform.